# Isolation, Structural Analysis and Biological Activity Assays of Biselisabethoxanes A and B: Two Dissymmetric *Bis*-Diterpenes from the Southwestern Caribbean Sea Gorgonian Coral *Pseudopterogorgia elisabethae*

**DOI:** 10.3390/molecules27227879

**Published:** 2022-11-15

**Authors:** Ileana I. Rodríguez, Abimael D. Rodríguez, Charles L. Barnes

**Affiliations:** 1Department of Chemistry, University of Puerto Rico–Río Piedras Campus, P.O. Box 23346, UPR Station, San Juan, PR 00931, USA; 2Molecular Sciences Research Center, University of Puerto Rico–Río Piedras Campus, 1390 Ponce de León Avenue, San Juan, PR 00926, USA; 3Department of Chemistry, University of Missouri-Columbia, Columbia, MO 65211, USA

**Keywords:** *Pseudopterogorgia elisabethae*, biselisabethoxanes, hetero-Diels-Alder reaction, *bis*-diterpenes, diterpene dimers, gorgonian octocoral

## Abstract

Two novel dissymmetric diterpenoids, biselisabethoxanes A and B (**1** and **2**), were isolated from the hexane extracts of the gorgonian coral *Pseudopterogorgia elisabethae*. Biselisabethoxane A (**1**) represents the first example of a marine-derived C_40_ dimer made of two distinct diterpene fragments, whereas biselisabethoxane B (**2**) is a fused heterodimer stemming from coupling of two amphilectane-based fragments. The structures of **1** and **2** were elucidated based on 1D and 2D NMR spectral data analysis. The molecular structure of **1** was subsequently confirmed by X-ray crystallographic analysis. When evaluated for their inhibitory effects in a series of well-established biological activity assays the isolated compounds were shown to moderately inhibit the growth of *Mycobacterium tuberculosis*.

## 1. Introduction

*Bis*-diterpenes (also known as diterpene dimers or tetraterpenes) are secondary metabolites consisting of two diterpene units bonded together by an amino, ether, or C–C bonds [1,2,3]. They constitute a class of compounds seldomly found among either terrestrial or marine-based organisms. A cursory search of the marine natural products chemistry literature reveals that this type of secondary metabolites is quite infrequently isolated from coelenterates. Accordingly, in the past few decades less than 25 representatives of the *bis*-diterpene family of compounds have been reported from this phylum [4].

Throughout the last two decades a handful of researchers have decidedly exposed the chemical versatility of the gorgonian octocoral *Pseudopterogorgia elisabethae* (Bayer, 1961) as an architect of bioactive natural products with novel carbon designs [5,6,7,8,9,10,11,12,13,14,15,16,17]. Likewise, our own laboratory has been an active participant in such bountiful an endeavor [18]. As part of this work, we now wish to report the isolation and structure elucidation of two dissymmetric C_40_ *bis*-terpenes isolated from this organism, namely, biselisabethoxane A (**1**) and biselisabethoxane B (**2**) (Figure 1) [19,20]. The first compound, **1**, represents a first-of-its-class *bis*-diterpene contrived of two distinctive skeletal classes of diterpenes, namely amphilectane and elisapterane, joined by an ether linkage. The second representative, **2**, can be envisioned as a novel *bis*-diterpene deriving from an inverse electron-demand hetero-Diels-Alder cycloaddition reaction comprising two units of the amphilectane-class of diterpenes [21,22].

## 2. Results and Discussion

Healthy colonies of *P. elisabethae* were collected by SCUBA off the San Andrés Archipelago, Colombia at depths of 80–100 ft in May 1996. The coral specimens were air-dried and kept frozen prior to homogenization. The dry animal (1.0 kg) was blended using a 1:1 mixture of MeOH–CHCl_3_ (11 × 1L) and, after filtration, the crude coral extract was concentrated under vacuum to yield an oily residue (284 g) that was subsequently suspended in H_2_O and partitioned against hexane. The resulting organic extract was concentrated in vacuo to afford a thick oily residue (178 g), a small portion of which (28%) was dissolved in toluene, filtered over sand and Celite, and then partitioned by size exclusion CC (2 × 25 g over Bio-Beads^®^ SX-3 in toluene). The terpene rich fractions were combined and later purified by successive silica gel CC with mixtures of EtOAc–hexane as eluent followed by reversed-phase ODS silica gel CC using mixtures of MeOH–H_2_O. Fractions were routinely pooled based on their TLC and NMR (^1^H and ^13^C) profiles. These iterative procedures led to comparable amounts of pure biselisabethoxane A (**1**, 11.0 mg, 0.02%) and biselisabethoxane B (**2**, 10.6 mg, 0.02%) suitable for structure elucidation work.

### 2.1. Structure Elucidation

The molecular structure of biselisabethoxane A (**1**), [α]^20^_D_ +123.1° (*c* 2.1, CHCl_3_), was proposed initially on the basis of comprehensive analysis of the 1D and 2D NMR (^13^C, ^1^H, ^1^H-^1^H COSY, HSQC, HMBC, and NOESY), IR, UV, and HR-FAB-MS spectra. A single-crystal X-ray structure analysis was subsequently carried out to confirm the proposed molecular structure. The molecular formula of **1** was established as C_40_H_54_O_5_ from HR-FAB-MS of an intense [M + K]^+^ pseudomolecular ion at *m*/*z* 653.3638 (calcd for C_40_H_54_O_5_K, 653.3608) revealing that **1** possessed 14 degrees of unsaturation. Seven of these were assigned to five C=C bonds (δ 147.1 (s, C-9), 146.9 (s, C-17′), 141.1 (s, C-2′), 137.2 (s, C-10), 135.9 (s, C-13), 131.4 (d, C-14), 129.6 (s, C-11), 129.6 (s, C-12), 128.3 (s, C-15), 126.2 (s, C-8)) and two C=O bonds (δ 203.1 (s, C-14′), 196.0 (s, C-17′)) showing that biselisabethoxane A was heptacyclic. IR absorptions at ν_max_ 3461(broad), 1760 (s), 1649 (s) and 1613 (s) cm^−1^ hinted at the presence of hydroxyl, carbonyl, and olefin functionalities. UV maxima at λ_max_ 210 (ε 61500), 234 (sh) and 289 (ε 10700) nm were consistent with the presence in **1** of benzene and enone functionalities. The appearance of an extra UV absorption (λ_max_ 345 nm) upon addition of 1 drop of 5% methanolic KOH confirmed the presence of a phenol group. Two D_2_O exchangeable protons (δ 5.97 (1H, br s) and 6.18 (1H, s)) in the ^1^HNMR spectrum implied the ubiquity in **1** of two hydroxyl groups.

The atypical complexity of the ^1^H and ^13^C NMR spectra, together with the molecular formula, early on suggested that biselisabethoxane A (**1**) was in fact a novel dissymetric *bis*-diterpene. This contention was supported by the detection of a myriad of closely-spaced (or overlapped) ^1^HNMR signals ascribable to 10 methyl groups, i.e., δ 1.96 (3H, s, Me-20), 1.70 (3H, br s, Me-17), 1.66 (6H, br s, Me-16, Me-20′), 1.25 (3H, d, *J* = 6.7 Hz, Me-19), 1.16 (3H, d, *J* = 7.1 Hz, Me-18′), 1.06 (3H, d, *J* = 6.4 Hz, Me-18) and 1.02 (9H, br s, Me-12′, Me-13′, Me-19′). Further support stems from HR-EI-MS data, which reveal that upon ionization and subsequent cleavage about the ether linkage, two complementary ion fragments are produced for C_20_H_27_O_3_^+^ and C_20_H_28_O_2_^•+^ (an even-electron ion at *m*/*z* 315.1951 along with an odd-electron ion at *m*/*z* 300.2089, respectively).

Closer inspections of the 1D (^1^H and ^13^C NMR) and 2D (^1^H–^1^H COSY, HMQC, and HMBC) NMR data (Appendix A) allowed us to promptly establish the structures of the two diterpene units comprising **1**. The presence of an amphilectane-based diterpene was quite evident on the basis of the vinylic methine doublet at δ 4.94 (1H, br d, *J* = 9.3 Hz, H-14), two broad methyl singlets at δ 1.70 (Me-17) and 1.66 (Me-16) and the conspicuous allylbenzylic methine at δ 3.60 (1H, dd, *J* = 8.7, 8.4 Hz, H-1) observed in the ^1^HNMR spectrum, ascribable to a tell-tale isobutenyl side chain attached to the C-1 position of the amphilectane ring of **1**. Moreover, there were two additional benzylic methine signals at δ 3.20 (m, H-7) and 2.01 (m, H-4) along with an aromatic methyl [δ 1.96 (s, Me-20)] consistent with the fully substituted benzene ring frequently found in many amphilectane-based diterpenes from *P. elisabethae* [23,24,25].

To satisfy the remaining degrees of unsaturation, we knew that the second diterpene unit in **1** had to be tetracyclic. The most salient structural features linked to the remaining half of the structure were: two ketone carbonyls (ν_max_ 1760 (s) cm^−1^ and 1649 (s) cm^−1^, assigned in the end, respectively, to cyclopentanone and 2-hydroxy-2-cyclohexenone moieties], an enol (δ_H_ 6.18 (1H, br s, exchangeable); δ_C_ 146.9 (s) and 141.1 (s)), a tertiary carbon bearing oxygen (δ_C_ 85.0 (s)), and quite a distinctive quaternary *sp*^3^ carbon (δ_C_ 61.6 (s)). These characteristic features were ominously reminiscent of the elisapterosins, a rare class of natural products isolated in the early 2000s from the same octocoral specimen [26]. Thus, side-by-side comparisons of the ^1^H and ^13^C NMR signals of authentic elisapterosin B (**3**) (Figure 2) with peaks stemming from the second diterpene unit of **1** allowed us to quickly unravel its structural identity as shown. Application of 2D NMR methods resulted in the unambiguous assignment of all protons and carbons as listed in Table 1 and allowed the complete planar structure for **1** to be assigned.

While segments of the relative stereochemistry within each of the diterpenoid halves in **1** were readily assigned by NOE NMR spectral methods (see Appendix A), some sectors had to be assigned indirectly based on the overall correlations observed in the NOESY spectrum (for instance, we could not confidently assign the relative configuration at C-1′ or C-15′ with the NMR data already at hand). Also, the proximity of several key signals in the ^1^H NMR spectrum of **1** interfered with these efforts. Thus, confirmation of the entire structure of biselisabethoxane A by single-crystal X-ray diffraction analysis was highly desirable. Recrystallization of **1** by slow evaporation of a concentrated MeOH/H_2_O solution gave crystals of excellent quality that were amenable to X-ray crystallographic analysis. The X-ray structure, which defines only the relative configuration, is shown in Figure 3. That notwithstanding, a comparison of the [α]_D_ values of **1** (+123.1°), **3** (–3.0°) and **4** (+111°) (Figure 1 and Figure 2) reveal that two of these compounds are strongly dextrorotatory implying that the optical rotation of **1** can be mostly attributed to the isolated amphilectane-based chiral unit. Since the structures shown for elisapterosin B (**3**) [26,27] and the *O*-benzyl ether derivative of pseudopterosins G-J aglycon (**4**) [28,29] outline their absolute configuration, these contentions further insinuate that structure **1** probably also depicts the absolute configuration of biselisabethoxane A (1*S*, 3*S*, 4*R*, 7*S*, 1′*S*, 3′*S*, 6′*R*, 7′*S*, 9′*S*, 10′*S*, 15′*S*).

Biselisabethoxane B (**2**) was isolated as an optically active bright yellow oil, [α]^20^_D_ +87.6° (*c* 1.3, CHCl_3_), with a molecular formula of C_40_H_52_O_4_ on the basis of HR-FAB-MS data: [M + Na]^+^ *m*/*z* 619.3788 (calcd for C_40_H_52_O_4_Na, 619.3763). UV maxima at λ_max_ 288 (ε 9700) and 314 (ε 13 000) nm imply that an extended π system exists in **2** over a longer series of atoms. The ^1^H and ^13^C NMR data for **2** are presented in Table 2. Unlike in *bis*-diterpene **1**, the ^13^C NMR spectrum of **2** showed 40 sharp resonance lines, 14 of which were ascribable to C=C double bonds (δ 157.9 (s, C-13′), 144.7 (s, C-1′), 137.8 (s, C-10), 137.2 (s, C-9), 132.5 (s, C-12′), 132.3 (s, C-13), 131.8 (d, C-14), 129.7 (s, C-12), 129.4 (s, C-15′), 129.3 (s, C-8′), 127.6 (s, C-15), 126.4 (d, C-14′), 126.3 (s, C-8), 122.3 (s, C-11)], a conjugated C=O double bond [δ 191.2 (s, C-9′)], a hemiketal carbon [δ 90.1 (s, C-10′)) and a tertiary carbon bearing an oxygen atom (δ 78.8 (s, C-11′)). The presence of hydroxyl and α,β-unsaturated carbonyl groups in **2** was deduced from the strong IR bands at ν_max_ 3460 (sharp) and 1664 (s) cm^−1^, respectively. The most conspicuous signals in the ^1^H NMR spectrum of **2** consisted of two olefinic methines (δ 5.71 (1H, br s, H-14′) and 5.00 (1H, dd, *J* = 10.2, 1.0 Hz, H-14)), a very sharp D_2_O exchangeable proton (δ 4.70 (1H, s, C-10′-OH)), an allylbenzylic methine (δ 3.65 (1H, dd, *J* = 8.8, 8.4 Hz, H-1)), two benzylic methines (δ 3.35 (1H, q, *J* = 7.7 Hz, H-7) and 2.82 (1H, q, *J* = 6.9 Hz, H-7′)), and ten well-discernable methyl resonances (δ 1.84 (s, Me-20), 1.76 (d, *J* = 1.0 Hz, Me-16′), 1.68 (d, *J* = 0.8 Hz, Me-17), 1.63 (d, *J* = 1.0 Hz, Me-16), 1.50 (d, *J* = 0.8 Hz, Me-17′), 1.45 (s, Me-20′), 1.22 (d, *J* = 6.6 Hz, Me-19′), 1.21 (d, *J* = 6.7 Hz, Me-19), 1.00 (d, *J* = 5.9 Hz, Me-18), 0.95 (d, *J* = 6.5 Hz, Me-18′)). Taken together, these data strongly suggested that biselisabethoxane B (**2**) was also a tetraterpene with 15 degrees of unsaturation. Like **1**, compound **2** therefore must be heptacyclic.

Interestingly, half of the carbon atoms attributed to the western hemisphere of **2** consisted of seven quaternary *sp*^2^ carbons (C-8–C-13 and C-15) plus three isolated methyl groups (C-16, C-17, and C-20). On the other hand, connectivities from C-1 to C-7 were inferred from the ^1^H-^1^H COSY cross-peaks, including correlations from H-1 to H-14, H-3 to H_3_-18 and H-7 to H_3_-19. This extended spin system, encompassing the second half of the carbon atoms within this unit, was quickly recognized as it was present in **1** and akin compounds isolated from the same gorgonian specimen (i.e., pseudopterosins G-J aglycon **4**) [28,29]. Confirmation of the proton connectivity network already established from the ^1^H–^1^H COSY and TOCSY experiments was obtained directly from long-range ^1^H–^13^C couplings. Additional HMBC correlations defining the structure of the C_20_H_26_O_2_ amphilectane-based subunit are depicted in Appendix A. Clearly, the chemical displacements of C-9, C-10, C-10′ and C-11′ (δ 137.2, 137.8, 90.1, and 78.8, respectively) suggested that the fully substituted benzene ring in **2** might be a part of a 2,3-dihydro-1,4-benzodioxine functionality (vide infra) [30,31,32,33]. Conversely, the structural elucidation of the remaining polycyclic terpenoid unit within **2**, despite having the same C_20_H_26_O_2_ composition, was more difficult to achieve as its spectroscopic data deviated from that of other diterpene congeners previously isolated from *P. elisabethae* [7,29].

While the ^1^H NMR signals ascribable to the eastern hemisphere of **2** (Table 2) looked conspicuously different, the spectrum revealed three singlet methyls (δ 1.45, 1.50, 1.76) and two doublet methyls (δ 0.95 and 1.22) along with an olefinic broad singlet resonance (δ 5.71, 1H), all of which pointed to a modified amphilectane skeleton. For instance, this time the familiar signals for an allylbenzylic methine near 3.65 ppm, the two benzylic methines, and the aromatic methyl group, were either missing or appeared noticeably shifted. Thus, connectivities from the ^1^H-^1^H COSY cross-peaks began with C-2′ and ended at C-7′ (including correlations from H-3′ to H_3_-18′ and H-7′ to H_3_-19′) indicating that the shortened spin system encompassed only eight *sp*^3^ carbon atoms. The other 12 carbons atoms about this unit were accounted for by ^13^C- and DEPT-135 NMR experiments as a carbonyl carbon, five quaternary *sp*^2^ carbons, an isolated *sp*^2^ methine, three isolated methyls, a hemiketal carbon and a tertiary carbon bearing oxygen. Proceeding under the assumption that the second diterpene unit in **2** was indeed another amphilectane-based ring, we assembled these carbon atom sets in a manner thoroughly consistent with all the ^1^H–^13^C long-range couplings observed in the HMBC spectrum (Appendix A). These data permitted the connection of the C_20_ moieties through two oxygen bridges encompassing either C-9/C-11′ and C-10/C-10′ or C-9/C-10′ and C-10/C-11′. Unfortunately, no HMBC correlations were found to establish the linkage of the two fragments because there are no hydrogen atoms at C-9, C-10, C-10′, and C-11′ in **2**. Thus, applying these combined 2D NMR methods resulted in the unambiguous assignment of all protons and carbons as listed in Table 2, and allowed two candidate planar structures for biselisabethoxane B to be assigned (**A** or **B**). As illustrated in Figure 1, the inferred tetraditerpenoid structures consisted of two amphilectane diterpenes (each of C_20_H_26_O_2_ composition and seven degrees of unsaturation) adjoined by a 2,3-dihydro-1,4-benzodioxine ring [30,31,32,33]. The latter functionality expanded the total index of hydrogen deficiency to 15 as required by the molecular formula of the natural product. Thus, the only difference between regioisomers **A** and **B** was the connection pattern of the two hemispheres.

Since no relevant heteronuclear correlation data could be detected between the two monomeric units, further data required for structural characterization were acquired while assessing the relative stereochemistry of **2** through supplementary 2D NMR techniques (Appendix A). Accordingly, the relative stereochemistry about the amphilectane rings in the natural product was found to be identical to that in **4** from analysis of the NOESY spectrum, coupling constant analysis, and comparisons of the NMR chemical shifts with those of known models. For instance, the 2-methyl-1-propenyl side chain at C-1 (located in the western hemisphere of regioisomers **A** and **B** in Figure 1) was confidently assigned with the α-orientation since H-14 resonated at δ 5.00 [28,29]. NOESY correlations of H-1/H-3 and H-4/H_3_-18 indicated that H-1 and H-3 must be pseudoaxial and that the isobutenyl side chain and H_3_-18 are both pseudoequatorial. A double triplet (observed in Bz-*d*_6_) at δ 2.25 with a large coupling constant (*J* = 10.1, 5.6 Hz) ascribable to H-4 suggested that the latter methine is *trans*-diaxial to H-3 and H-5β. Additional NOESY correlations of H-2α/H-4, H-4/H-5α, H-5α/H-6α, H-6α/H_3_-19, and H-6β/H-7 revealed that H-4 and the methyl group at C-7 are in a *cis* 1,4-pseudodiaxial conformation. Hence, the isobutenyl group at C-1, the methyl groups at C-3 and C-7, and H-4 are all α-oriented. By the same token, the relative stereochemistry of the methyl groups at C-3′ and C-7′, and H-4′ alongside the opposite hemisphere was also established to be all α-oriented. On the other hand, to determine which constitutional framework, **A** or **B**, depicts the correct structure for the natural product, we meticulously searched for NOE cross-peaks stretching across the two amphilectanoid hemispheres. First, a faint but discernible NOE interaction between the C-10′ hydroxyl proton at δ 4.70 and its neighboring H_3_-20′ angular methyl at δ 1.45 established their *cis* orientation. Furthermore, strong NOE correlations observed between H_3_-20′ with H_3_-19 could only be explained on the assumption that the natural product has the **A** constitution (Figure 1) and that the angular –OH and methyl substituents at C-10′ and C-11′, respectively, are both α-oriented. In support of this assignment pivotal NOE’s were observed which placed H-7 near H-14′ and H_3_-16′. These correlations, together with the ^1^H and ^13^C NMR data (recorded in both CDCl_3_ and Bz-*d*_6_) and compelling UV-visible absorption properties, led us to conclude that **2** is the most likely molecular structure for biselisabethoxane B. In all, four possible structures were proposed by the rational linkage of the two amphilectane fragments. Aside from **2**, we also evaluated two alternate stereoisomers bearing the hydroxyl and H_3_-20′ groups *trans* and a third one in which the groups were *cis* yet β-oriented. However, simulations of these alternative stereostructures predicted strong NOE interactions that were generally not detected in the NOESY spectrum of **2**. Conversely, the energy-minimized models predicted the occurrence of protons with inter-nuclear distances exceeding 3.2 Å for which diagnostic NOE correlations were in fact detected between them. Strategic bond disconnections about the 2,3-dihydro-1,4-benzodioxine moiety led us to envision biselisabethoxane B (**2**) as a likely hetero-Diels-Alder adduct arising from dimerization of *ortho*-benzoquinone **7** under conditions that would facilitate proton transfer and hence tautomerization (Figure 2) [34]. The latter compound could stem from oxidation of biogenetic precursor pseudopterosin G-J aglycon (**4**) of known absolute configuration. Coincidentally, the known diterpenes elisabatin A (**5**) and elisabatin B (**6**) (Figure 2) obtained from the same extracts as **1** and **2**, suggest comparable functionalities as those brought forth by **7** and **7a** in Figure 2 [35,36]. Alternatively, formation of intermediate **7a** might ensue from plain dehydration of elisabethol (**8**) (Figure 2), a structurally related diterpene reported by Kerr et. al. from South Floridian specimens of *P. elisabethae* [10]. As a matter of interest, the crystal structure of **6** has been shown to consist of two centrosymmetrically related, H-bonded molecules of elisabethin B. Thus, elisabethin B forms discrete dimers by a pair of H-bonds between the carbonyl and hydroxyl groups of the molecules related by the center of symmetry (Figure 4). Accordingly, we contend that tautomers **7** and **7a** orient themselves in a similar fashion (i.e., the unparalleled alignment of the two amphilectane fragments may benefit from an internal H-bond between the C-10 carbonyl and the C-10′ hydroxyl groups) prior to undergoing cycloaddition to afford the most sTable 1,4-product (**A** in Figure 1). Cautious extrapolation of these results suggests that in the case of **7** similar interactions in the solid state might cause the *ortho*-benzoquinone and its tautomer to assume a fixed spatial orientation that controls the regiospecificity of the Hetero-Diels-Alder reaction.

Although preorganization of the Diels-Alder components is not expected to exist in solution, in water (i.e., the gorgonian’s habitat) cycloaddition must benefit from these molecules being forced closely together to mimic the situation in the solid state [36]. To contend the origin of the purported regioselectivity of the hetero-Diels–Alder product **2**, a simple charge distribution was calculated (Appendix A). The study revealed that in *ortho*-benzoquinone **7** there was no significant difference between the electron density of the carbonyl oxygens at C-9 and C-10. To understand the reactivity parameters of the above systems, we carried out theoretical calculations using the Spartan’18 Program (version 1.4.6). The results are given in Appendix A. From the HOMO-LUMO energy differences it is evident that the proposed reaction proceeds by inverse electron demand. Therefore, it is possible that a steric effect of **7** and **7a** in the transition state could be a main factor in controlling the regioselectivity of the dimerization, giving compound **2** as the only product via a less sterically demanding route. It is noteworthy here that even after a long-standing chemical investigation of *P. elisabethae*, we never detected the presence of the complementary 1,3-product (regioisomer **B** in Figure 1) nor have we encountered C_40_ *bis*-diterpenes stemming from self-same dimerization of **7** (see Appendix A). As per Figure 2, biselisabethoxane B (**2**) can only be formed if *ortho*-benzoquinone **7** tautomerizes. Since the latter precursor is fully substituted and hence quite bulky, a more sterically demanding transition state will ensue, thus precluding the self-same dimerization route.

We surmise that the specific rotation of biselisabethoxane B, [α]^20^_D_ +87.6°, is set exclusively by the dominant influence of the chiral amphilectane units of well-defined configuration with little to no contributions to [α]_D_ stemming from the 2,3-dihydro-1,4-benzodioxine system. It is reasonable, therefore, to conclude from the specific rotation and nearly identical ^1^H and ^13^C chemical shifts about the intact western hemisphere ring, being comparable to those of **1** and **4** [28,29], that the most likely absolute configuration for biselisabethoxane B (**2**) is 1*S*, 3*S*, 4*R*, 7*S*, 3′*S*, 4′*R*, 7′*S*, 10′*S*, 11′*S*.

### 2.2. Biological Evaluation of the New C_40_ Bis-Diterpenes

Anti-infective activity. When tested for their inhibitory activity toward the growth of *Mycobacterium tuberculosis* H_37_Rv at a single concentration (6.25 μg/mL), *bis*-diterpene **1** exhibited the strongest inhibitory activity (24%), whereas *bis*-diterpene **2** reached a comparable level of potency (25%) only at ~ ten times the concentration (64 μM). Interestingly, we had previously reported that elisapterosin B (**3**) was found to effect strong inhibitory activity (79%) against *M. tuberculosis* H_37_Rv at a concentration of 12.5 μg/mL [26]. These data suggest that the superior antitubercular properties of *bis*-diterpene **1**, when compared to those of **2**, most certainly stem from its C_20_ elisapterane-based motif. When screened for anti-plasmodial activity against the *Plasmodium falciparum* W2 (chloroquine-resistant) strain compounds **1** and **2** compounds were deemed inactive as their respective IC_50_ values were found to be >50 μM.

Cytotoxic activity. Compounds **1** and **2** were evaluated in a three-cell line panel at a single high dose (10 M) consisting of MCF-7 (breast cancer), NCI-H460 (non-small-cell lung cancer), and SF-268 (CNS) cells. Results from the one dose primary anticancer assay indicated that each compound lacked significant cytotoxicity. As neither compound satisfied pre-determined threshold inhibition criteria in a minimum number of cell lines, *bis*-diterpenes **1** and **2** were not recommended by the NCI for progression to the full five-dose assay.

Neurodegenerative and neuroinflammation activity. Considering the potential ability of secondary metabolites from *P. elisabethae* to inhibit inflammation, we screened *bis*-diterpenes **1** and **2** for neurotoxic and neuroinflammatory activity [5,6]. As activation of brain microglia and concomitant release of both O_2_^−^ and TXB_2_ have been reported in neurodegenerative disorders and neuroinflammation, we also looked at the possible concentration-dependent effect of *bis*-diterpenes **1** and **2** on the release of O_2_^−^, TXB_2_, and lactate dehydrogenase (LDH), a marker for cell toxicity. Surprisingly, both diterpene dimers demonstrated minimal effects, if any, on TBX_2_, O_2_^−^ and LDH release.

Cyclin-dependent kinases inhibitory activity. CDKs are a family of protein kinases the activity of which has been shown to be required for initiation and traverse of specific phases of the cell cycle as well as regulation of transcription. As several new small molecules have been reported as having the capacity to target and inhibit cyclin B kinase (CNRS), compounds **1** and **2** were also screened for the inhibition of cyclin B kinase [37]. Sadly, no significant inhibitory effect against this protein kinase was detected. Clearly, compounds **1** and **2** are structurally different than previously described CDK inhibitors as they could not target the CDK/cyclin B binding site.

Antiviral activity. Only the more abundant of the two compounds, biselisabethoxane A (**1**), was subjected to anti-viral testing against three different classes of viruses (Herpex Simplex virus (HSV-1), Hepatitis B virus (HBV), and Influenza A virus (FLU-A)) as well as to a protease inhibition screening. Acyclovir (ACV) was used as a control. The tests showed conclusively that *bis*-diterpene **1** was not significantly active against each virus.

## 3. Materials and Methods

### 3.1. General Experimental Procedures

Optical rotations were measured in CHCl_3_ at 589 nm with a Rudolph Autopol IV automatic polarimeter using a 10 cm microcell. IR and UV spectra were measured with a Nicolet Magna FT-IR 750 spectrophotometer and a Shimadzu UV-vis spectrometer (UV-2401PC), respectively. ^1^H and ^13^C NMR spectra were recorded in CDCl_3_ at 500 and 125 MHz, respectively, with a Bruker Avance DRX-500 spectrometer with TMS as internal standard. Multiplicities in the ^1^H NMR spectra are described as follows: s = singlet, d = doublet, t = triplet, q = quartet, m = multiplet, dd = double doublet, and br = broad, and coupling constants are reported in Hertz. 2D-NMR experiments (^1^H–^1^H COSY, NOESY, DEPT-135, HMQC, and HMBC) were also measured with the same instrument. HR-EI-MS and HR-FAB-MS data were determined using a VG Micromass AutoSpec (Fisons) magnetic sector mass spectrometer (70 eV). Silica gel (35–75 mesh) or bonded C-18 silica gel (35–75 mesh), Bio-Beads SX-3 were used for column chromatography, and precoated silica gel GF_254_ plates were used for TLC and were revealed by exposure to I_2_ vapors or heating the plates sprayed with 5% H_2_SO_4_ in EtOH. All solvents used were either spectral grade or were distilled from glass prior to use. Commercially available reagents were purchased from Sigma–Aldrich (Saint Louis, MO, USA) and used as received unless stated otherwise. The percentage yield of each compound is based on the weight of the dry gorgonian MeOH–CHCl_3_ extract.

### 3.2. Biological Material

The biological specimens used during this investigation corresponded to a deep-water morphotype of *Pseudopterogorgia elisabethae* Bayer (order Gorgonacea, family Gorgoniidae, phylum Cnidaria) collected in May 1996 by scuba from deep reef waters (–28 m) off San Andrés island, Colombia (located at 12°33′ N 81°43′ W). A voucher specimen (No. PESAI-01) has been deposited at the Chemistry Department of the University of Puerto Rico.

### 3.3. Collection, Extraction, and Isolation of Bis-Diterpenes 1 and 2

Colonies of *Pseudopterogorgia elisabethae* were collected by hand using SCUBA at depths of –80 to –100 ft off San Andrés Island, Colombia. The gorgonian was air-dried, lyophilized and kept frozen prior to homogenization. The dry animal (1.0 kg) was blended in CHCl_3_–MeOH (1:1) (11 × 1 L) and, after filtration, the crude extract was evaporated under vacuum to yield 284 g of a green residue. The crude extract was then suspended in H_2_O and partitioned with hexane, CHCl_3_, EtOAc, and *n*-butanol. The resulting hexane extract was concentrated in vacuo to yield an oily residue (178 g). The latter extract was divided in two portions for further analysis (portion A: 50 g and portion B: 128 g). The organic extract in portion A was diluted in a small volume of toluene, filtered over sand and Celite^®^, and then partitioned by size exclusion chromatography (2 × 25 g over Bio-Beads^®^ SX-3 in toluene). Four primary fractions, denoted PEH 1 (24.1 g), PEH 2 (9.2 g), PEH 3 (15.1 g), and PEH 4 (1.6 g), were obtained. The terpenoid-rich fraction PEH 2 was further purified by CC over normal phase silica gel (275 g) using a gradient solvent system (3–100% EtOAc in hexane). This led to the isolation of 29 subfractions labeled PEH 2.1–PEH 2.29. Fraction PEH 2.11 (147.1 mg) was separated further by reversed-phase CC over 5 g ODS-silica upon eluting with a 90:10 CH_3_OH–H_2_O mixture. The sixth fraction eluted (52 mg) was purified further by CC over silica gel (3.0 g) with 98:2 hexane–EtOAc affording pure biselisabethoxane A (**1**, 11.0 mg, 0.02%). Consecutive fractionation of PEH 2.24 (591.0 mg), first by CC over silica gel (20 g) using 85:15 hexane–CHCl_3_ followed by CC over silica gel (15 g) with 99:1 hexane–EtOAc, led to an intense yellow-colored subfraction (54.9 mg). The latter material was purified further over reversed-phase ODS silica gel (3.0 g) using a 9:1 mixture MeOH–H_2_O to yield pure biselisabethoxane B (**2**, 10.6 mg, 0.02%).

Biselisabethoxane A (**1**): ivory crystals; [α]^20^_D_ +123.1° (*c* 2.1, CHCl_3_); IR (film) 3361 (broad), 2949 (s), 2924 (s), 2868 (s), 1760 (s), 1649 (s), 1613 (s), 1452 (s), 1388 (s), 1120 (s) cm^−1^; UV (MeOH) λ_max_ 234 (sh), 289 (ε 10 700), 303 (sh) nm; ^1^H NMR (500 MHz, CDCl_3_) and ^13^C NMR (125 MHz, CDCl_3_) see Table 1; EI-MS *m*/*z* [M]^+^ 614 (0.5), 315 (3), 300 (17), 244 (100), 229 (20), 206 (15); HR-EI-MS *m*/*z* 614.3924 (calcd for C_40_H_54_O_5_, 614.3971), 315.1951 (calcd for C_20_H_27_O_3_, 315.1960), 300.2099 (calcd for C_20_H_28_O_2_, 300.2089); HR-FAB-MS *m*/*z* 653.3638 [M + K]^+^ (calcd for C_40_H_54_O_5_K, 653.3608).

Biselisabethoxane B (**2**): bright yellow oil; [α]^20^_D_ +87.6° (*c* 1.3, CHCl_3_); IR (film) 3460 (broad), 2967 (s), 2949 (s), 2917 (s), 2855 (s), 2819 (s), 1664 (s), 1599 (m), 1552 (m), 1455 (s), 1445 (s), 1426 (s), 1374 (m), 1323 (m), 1202 (m), 1131 (m), 1069 (m) cm^−1^; UV-VIS (MeOH) λ_max_ 288 (ε 9700), 314 (ε 13 000) nm; ^1^H NMR (500 MHz, CDCl_3_) and ^13^C NMR (125 MHz, CDCl_3_) see Table 2; HR-FAB-MS *m*/*z* 619.3788 [M + Na]^+^ (calcd for C_40_H_52_O_4_Na, 619.3763).

### 3.4. Biological Assays

The primary in vitro inhibition bioassays used during this work were used as described before [29].

### 3.5. X-ray Crystallographic Data for Biselisabethoxane A (**1**)

Crystallization of biselisabethoxane A (**1**) by slow evaporation from a MeOH–H_2_O mixture yielded crystals of excellent quality. The X-ray data were collected on a Siemens SMART CCD system at 173(2) K. Crystal data: C_40_H_54_O_5_, *M_r_* = 614.8, monoclinic, space group *P2~1*~*, a* = 12.1072(6), *b* = 10.5771(6), *c* = 27.4452(14) Å, *V* = 3497.5(3) Å^3^, *Z* = 8, *ρ*_calc_ = 1.168 Mgm^−3^, *F*_000_ = 1336, *λ* (Mo_Kα_) = 0.71073 Å, *μ* = 0.075 mm^−1^. Data collection and reduction: crystal size, 0.50 × 0.25 × 0.05 mm^3^, *θ* range, 1.69–27.13°, 22,255 reflections collected, 8141 independent reflections (*R*_int_= 0.0757), final *R* indices (*I* > 2*σ(I)*): *R*_1_ = 0.0440, *wR*_2_ = 0.0689 for 848 variable parameters, GOF = 0.847. The crystallographic data for **1** reported in this article have been deposited at the Cambridge Crystallographic Data Centre under reference No. CCDC 678943. Copies of the data can be obtained, free of charge, on application to the Director, CCDC, 12 Union Road, Cambridge CB2 1EZ, UK (fax +44-1223-336,033 or e-mail deposit@ccdc.cam.ac.uk).

## 4. Conclusions

Two new C_40_ *bis*-diterpenoids, biselisabethoxanes A and B (**1** and **2**), were isolated from hexane extracts of *P. elisabethae*. The structures were fully characterized through spectroscopic methods, and that of **1** was later confirmed by X-ray crystallographic analysis. The condensation of two diterpenes forming a C_40_ *bis*-diterpenoid structure is quite rare in nature even if a few such compounds have already been reported from Caribbean gorgonian octocorals. Biselisabethoxane A (**1**) represents the first example of a dissymmetric C_40_ dimer formed by linking together two diterpenes of the amphilectane- and elisapterane-class through an ether bond. Our proposed biosynthesis suggests that biselisabethoxane B (**2**) was formed through a hetero-Diels–Alder reaction of *ortho*-benzoquinone **7** and tautomer **7a**. Heterodimer **2** is constituted by two amphilectanoid moieties and, due to the high reactivity of *ortho*-benzoquinones such as **7**, we cannot exclude the possibility that **2** is an artifact formed during the isolation process. However, as *ortho*-benzoquinone **7** was accessible to us by oxidation of *ortho*-catechol **4** through a known protocol [23,29], we tested this contention and sought to obtain **2** from **7** under a variety of reaction conditions. The fact that we never observed the formation of **2** during these attempts hints at the notion that biselisabethoxane B (**2**) does occur naturally. While the corpus of bioactivity information available for compounds **1** and **2** is thus far insufficient to assess their biological relevance, the abundance of unique features endowing their molecular structures makes them attractive subjects for further biological scrutiny.

## Data Availability

The data presented in this study are available in this article.

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
