# Peer review of "Isolation, Structural Analysis and Biological Activity Assays of Biselisabethoxanes A and B: Two Dissymmetric Bis-Diterpenes from the Southwestern Caribbean Sea Gorgonian Coral Pseudopterogorgia elisabethae"

_molecules, 2022, doi:10.3390/molecules27227879_

Round 1
Reviewer 1 Report
The manuscript by Ileana I. Rodríguez, Abimael D. Rodrígueza and Charles L. Barnes « Isolation, Structural Analysis and Biological Activity Assays of Biselisabethoxanes A and B: Two Dissymmetric Bis-Diterpenes from the Southwestern Caribbean Sea Gorgonian Coral Pseu dopterogorgia elisabethae» is devoted to isolation and structure elucidation of two novel unique diterpenoids. The structures were elucidated not only using NMR spectroscopy, but also X-ray crystallographic analysis. These compounds were shown to inhibit the growth of Mycobacterium tuberculosis. The manuscript is written in good language and can be publushed in Molecules.
Author Response
Response to Reviewer 1: Other than the check marks shown in the "Open Review" section no specific suggestions/comments for improvement were provided by this reviewer.
Reviewer 2 Report
The manuscript “Isolation, Structural Analysis and Biological Activity Assays of Biselisabethoxanes A and B: Two Dissymmetric Bis-Diterpenes from the Southwestern Caribbean Sea Gorgonian Coral Pseudopterogorgia elisabethae" is devoted to the isolation and characterization of diterpenoids from gorgonian coral. UV-Vis, FTIR, NMR spectroscopies were used for characterization of obtained substances. Structure elucidation is quite thorough. Biological activity is much less discussed, and in general I would like to see more data on this topic. The results obtained can be useful from the point of view of biochemistry and physiology.
I think, this manuscript can be published in the Molecules after minor revision taking into account general recommendation and some of the remarks described below:
- Abstract: It would be better to add some general conclusion about biological activity in the end of the Abstract.
- Provide, please, more spectral data (FTIR figures, UV-Vis figures) in the manuscript.
- Provide, please, more data discussion devoted to the biological activity.
- It is necessary to make a full band assignment for the FT-IR data. You also need to specify the intensity of each specified band.
-
Too much self-citation for the author A.D. Rodriguez. His name is mentioned in articles 5, 6, 8, 11, 25, 26, 27, 28, 29, 30, 31, 37, 40, 46. The level of self-citation reaches 30% (14 from 47) with a maximum of 15-20%.
- Years of equipment production must be given. The reagent manufacturer must be brought in.
Author Response
The following changes were made upon following the comments/suggestions received from Reviewer 2:
(1) At the end of the Abstract, we added a sentence providing a general conclusion about the biological activity.
(2) To provide more data three figures provided originally as Supplementary Material (Figures S1–S3) are now inserted into the manuscript.
(3) More discussion (including data) devoted to the subject of biological activity was added in that section of the manuscript. Also, at the end of the conclusions, we inserted an extra sentence on this same subject.
(4) The intensity of all the FT-IR bands listed throughout the manuscript was specified. However, because the interpretability of IR spectra is always very low, we fully assigned only the relevant IR bands that are strong in intensity.
(5) To minimize self-citation 8 references from the author AD Rodriguez were removed (namely: 6, 8, 11, 25, 26, 27, 28, and 29). The rest of the references were left as they are considered essential to the discussion.
(6) The reagent manufacturer is now indicated in the Experimental section. On the other hand, we did not understand the following suggestion from this reviewer: "Years of equipment production must be given". We checked three recent articles from Molecules (all from 2022) and did not find this kind of information included in the Experimental section. Ordinarily, in our field, this kind of information is not requested for scientific publications.